# Plasma Sphingolipid Profile in Association with Incident Metabolic Syndrome in a Chinese Population-Based Cohort Study

**DOI:** 10.3390/nu13072263

**Published:** 2021-06-30

**Authors:** Huan Yun, Qi-Bin Qi, Geng Zong, Qing-Qing Wu, Zhen-Hua Niu, Shuang-Shuang Chen, Huai-Xing Li, Liang Sun, Rong Zeng, Xu Lin

**Affiliations:** 1Shanghai Institute of Nutrition and Health, University of Chinese Academy of Sciences, Chinese Academy of Sciences, Shanghai 200031, China; yunhuan2017@sibs.ac.cn (H.Y.); zonggeng@sibs.ac.cn (G.Z.); zhniu@sibs.ac.cn (Z.-H.N.); chenshuangshuang2017@sibs.ac.cn (S.-S.C.); lihx@sibs.ac.cn (H.-X.L.); sunliang@sibs.ac.cn (L.S.); 2Department of Epidemiology and Population Health, Albert Einstein College of Medicine, Bronx, NY 10461, USA; qibin.qi@einstein.yu.edu; 3CAS Key Laboratory of Systems Biology, Shanghai Institute of Biochemistry and Cell Biology, Center for Excellence in Molecular Cell Science, Chinese Academy of Sciences, Shanghai 200031, China; qqwu@sibs.ac.cn (Q.-Q.W.); zr@sibs.ac.cn (R.Z.); 4Key Laboratory of Systems Health Science of Zhejiang Province, Hangzhou Institute for Advanced Study, University of Chinese Academy of Sciences, Chinese Academy of Sciences, Hangzhou 310024, China

**Keywords:** ceramide, hydroxysphingomyelin, inflammation, metabolic syndrome

## Abstract

Although bioactive sphingolipids have been shown to regulate cardiometabolic homeostasis and inflammatory signaling pathways in rodents, population-based longitudinal studies of relationships between sphingolipids and onset of metabolic syndrome (MetS) are sparse. We aimed to determine associations of circulating sphingolipids with inflammatory markers, adipokines, and incidence of MetS. Among 1242 Chinese people aged 50–70 years who completed the 6-year resurvey, 76 baseline plasma sphingolipids were quantified by high-throughput liquid chromatography-tandem mass spectrometry. There were 431 incident MetS cases at 6-year revisit. After multivariable adjustment including lifestyle characteristics and BMI, 21 sphingolipids mainly from ceramide and hydroxysphingomyelin subclasses were significantly associated with incident MetS. Meanwhile, the baseline ceramide score was positively associated (RR_Q4 versus Q1_ = 1.31; 95% CI 1.05, 1.63; *p*_trend_ = 0.010) and the hydroxysphingomyelin score was inversely associated (RR_Q4 versus Q1_ = 0.60; 95% CI 0.45, 0.79; *p*_trend_ < 0.001) with incident MetS. When further controlling for clinical lipids, both associations were attenuated but remained significant. Comparing extreme quartiles, RRs (95% CIs) of MetS risk were 1.34 (95% CI 1.06, 1.70; *p*_trend_ = 0.010) for ceramide score and 0.71 (95% CI 0.51, 0.97; *p*_trend_ = 0.018) for hydroxysphingomyelin score, respectively. Furthermore, a stronger association between ceramide score and incidence of MetS was evidenced in those having higher inflammation levels (RR_Q4 versus Q1_ 1.57; 95% CI 1.16, 2.12; *p*_interaction_ = 0.004). Our data suggested that elevated ceramide concentrations were associated with a higher MetS risk, whereas raised hydroxysphingomyelin levels were associated with a lower MetS risk beyond traditional clinical lipids.

## 1. Introduction

In the past few decades, excessive calorie intake and an unhealthy sedentary lifestyle have driven a global epidemic of cardiovascular diseases (CVD) and type 2 diabetes (T2D), imposing enormous burdens of mortality and disability [1]. Metabolic syndrome (MetS), a cluster of multiple metabolic disturbances, including central obesity, dyslipidemia, hypertension, and hyperglycemia, is associated with two to five times heightened risks of CVD and T2D [2]. Identifying novel and potentially modifiable factors of MetS is critical for early prevention of more severe cardiometabolic diseases.

Clinical measures of blood lipids may not decipher the complexity of altered lipid metabolism and related mechanisms associated with onset of MetS [3]. Among the myriad lipid species, sphingolipids are a group of lipids with a sphingoid backbone and have emerged as signaling molecules that are critical players in various metabolic abnormalities [4]. A large body of evidence suggests that ceramides are particularly detrimental in metabolic health because they disrupt insulin action, vascular reactivity, and mitochondrial metabolism [5]. Inhibition of ceramide generation in vivo significantly improved glucose homeostasis and lipid metabolism [4]. Unlike ceramides, administration of sphingomyelins (SMs) in rats prevented dyslipidemia via suppressing intestinal absorption of cholesterol, fatty acids, and triglycerides [6].

Most published studies have reported that high concentrations of ceramides and SMs are strongly associated with increased CVD [7,8,9] and T2D risks [10,11,12], but the associations linking sphingolipids and risk of MetS remain to be clarified. To date, only a few cross-sectional and case–control studies with limited species have observed positive correlations between ceramides and MetS [13,14,15]. Notably, sphingolipids with distinct chain lengths and numbers of double bonds may impact cardiometabolic outcomes differently [16]. In our prior study, we observed that monounsaturated ceramides (d18:1/18:1, d18:1/20:1, d18:1/22:1) showed more pronounced associations with risk of T2D than saturated ceramide species in Chinese [17]. However, it remains unknown whether distinct sphingolipid species or classes show different associations with MetS.

Adverse profiles of inflammatory markers and adipokines have been documented to increase MetS risk in others’ and our previous studies [18,19]. Experimental studies revealed that inflammation stimuli promoted ceramide generation, whereas adiponectin suppressed cellular ceramide concentrations by enhancing the ceramidase activity [20]. Further, in animal studies, long-chain SMs supplementation was found to activate adiponectin signaling, as well as to diminish both acute and chronic inflammatory responses [6]. Thus, it is of interest to know whether and to what extent that inflammation status and adipokines could affect the sphingolipid-MetS associations.

To fill these knowledge gaps, the aims of our study were to estimate the associations of circulating sphingolipids individually and collectively with risk of incident MetS and to explore potential modifications of inflammatory markers and adipokines on these associations in an established 6-year cohort of middle-aged and elderly Chinese adults.

## 2. Materials and Methods

### 2.1. Study Population

The participants were from a population-based cohort entitled the Nutrition and Health of Aging People in China (NHAPC) study, which was designed to investigate environmental and genetic factors and their interactions on cardiometabolic diseases [21,22]. Briefly, 3289 adults aged 50–70 years from Beijing and Shanghai were recruited in 2005, and 2529 (76.9%) of them were followed for 6 years. When analyzing the RR of MetS, participants were excluded if they met any of the following criteria: (1) lacking baseline lipidomics data (*n* = 281); (2) having MetS or T2D at baseline (*n* = 1000); and/or (3) insufficient data to define incident MetS (*n* = 6). Therefore, 1242 participants were included for incidence of MetS (Appendix A). Additionally, when incidence of a specific MetS component was analyzed, participants with corresponding component at baseline were excluded. The numbers of participants in each component analyses were 1156 for central obesity, 1713 for hypertriglyceridemia, 1312 for low HDL-cholesterol, 702 for elevated blood pressure, and 1345 for hyperglycemia. The study protocol was approved by the Institutional Review Board of the Institute for Nutritional Sciences, and each participant provided a written informed consent in baseline and follow-up surveys.

### 2.2. Data Collection

Information on demography, lifestyle factors, medication use, and health status was collected by a standardized questionnaire during a face-to-face interview. Educational attainment (0–6 years, 7–9 years, or ≥10 years), current smoking (yes or no), current alcohol drinking (yes or no), physical activity (low, moderate, or high), family history of chronic diseases (yes or no), and use of lipid-lowering medication (yes or no) were previously defined [22]. All participants were required to fast overnight before physical examination. Body weight, height, waist circumference, and blood pressure were measured by trained staff following standard protocols [21]. Body mass index (BMI) was calculated as weight (kg) divided by the square of height (m^2^).

### 2.3. Laboratory Measurements

Fasting venous blood samples were collected by using EDTA-containing vacuum tubes, centrifuged at 4 °C, and stored at –80 °C until analyses, in baseline and follow-up surveys. Plasma fasting glucose, total cholesterol, LDL-cholesterol, HDL-cholesterol, and triglycerides were measured on an automatic analyzer (Hitachi 7080; Hitachi, Tokyo, Japan). Insulin level was detected by radioimmunoassay (Linco Research, St Charles, MO, USA) [18]. The measurements of baseline plasma C-reactive protein (CRP) [21], interleukin-6 (IL-6) [23], adiponectin [24], and retinol-binding protein 4 (RBP4) [25] were described previously. Homeostatic model assessment of insulin resistance (HOMA-IR) was calculated as insulin (μU/mL) × glucose (mmol/L)/22.5.

### 2.4. Sphingolipid Profile Quantification

Baseline plasma lipids were measured by targeted liquid chromatography-tandem mass spectrometry. Briefly, lipids were extracted following a modified methyl tert-butyl ether protocol. Liquid chromatographic separation was performed on Shimadzu Nexera X2 LC-30AD system, coupled with SCIEX 5500 QTRAP Mass Spectrometer. Analyst 1.6.3 software (Sciex, Foster City, CA, USA) was used for data acquirement. Waters ACQUITY UPLC BEH HILIC Column (130 Å, 1.7 µm, 2.1 mm × 100 mm) was applied for chromatographic separation. Plasma samples were analyzed in random order, with quality control samples inserted every 10 samples to ensure reliability of the lipidomic analysis. Finally, 728 lipid species were quantified. Detailed method of lipidomic analysis was reported previously [17].

In the current study, 76 sphingolipids were included: 12 ceramides, 9 dihydroceramides (dhCers), 43 SMs, and 12 glycosphingolipids (GSLs). Eight out of the 76 sphingolipids had missing values (missing rate < 0.1%) and the median coefficient of variation was 19.3% (range: 5.3–29.9%). Given the diverse chemical structures, SM species were categorized into SMs and hydroxysphingomyelins. Hydroxysphingomyelins were further classified as follows: (1) SM (OH)s, SMs with one additional hydroxyl; and (2) SM (2OH)s, SMs with two additional hydroxyls.

### 2.5. Assessment of Incident MetS

MetS was defined according to the updated National Cholesterol Education Program Adult Treatment Panel III criteria for Asian Americans [26]. Incident MetS was identified if participants presented three or more of five features in the 6-year revisit: (1) waist circumference ≥ 90 cm in men or ≥80 cm in women; (2) triglycerides ≥ 1.7 mmol/L; (3) HDL-cholesterol ≤ 1.03 mmol/L in men or ≤1.30 in women; (4) blood pressure ≥ 130/85 mmHg or currently taking any antihypertensive medication; and (5) fasting glucose ≥ 5.6 mmol/L, or taking any antidiabetic medications, or previous diagnosis of T2D.

### 2.6. Statistical Analysis

The missing values, which were below the detection limit, were imputed by half of minimum values. Baseline sphingolipid concentrations (mg/L) were log-transformed and standardized (mean = 0, SD = 1). Baseline characteristics were compared using ANCOVA for continuous variables and logistic regression for categorial variables between incident MetS cases and noncases. Partial Spearman correlation analysis was applied to examine correlations of sphingolipids with metabolic risk factors, inflammatory markers, and adipokines at baseline, after adjusting for age, sex, region (Beijing or Shanghai), and residence (urban or rural). The inflammatory index and adipokine index were computed as follows: (CRP z score + IL-6 z score)/2 and (adiponectin z score + RBP4 z score)/2, respectively [26]. The associations of baseline sphingolipids with risk of incident MetS and its components were examined in the following steps.

First, we conducted an exploratory principal component analysis (PCA) among 76 sphingolipid species. Factors with an eigenvalue > 2 were retained (Appendix A). Individual species with absolute loading >0.40 were considered as a relevant component in each factor. Due to the high incidence of MetS (34.7%) in our cohort, log-Poisson model was used to estimate RR (95% CI) for MetS and its components according to quartile and per SD increment of identified factors [27]. Model 1 was adjusted for age, sex, region, residence, educational attainment, current smoking, current alcohol drinking, physical activity, family history of chronic diseases, use of lipid-lowering medication, and BMI; model 2 was further adjusted for baseline triglycerides, LDL-cholesterol, and HDL-cholesterol; and model 3 was further adjusted for baseline HOMA-IR, inflammatory markers (CRP and IL-6), and adipokines (adiponectin and RBP4) to examine whether the observed association could be explained by these biomarkers as suggested based upon existing literatures [6,20].

Second, sphingolipid scores were further constructed among the main lipids represented in those PCA factors associated with incident MetS risk. Five scores, including ceramide score, dhCer score, SM score, SM (OH) score, and GSL score, were computed as the sum of standardized sphingolipid values [28]. Unlike the factor analyses, in the sphingolipid scores, both the known chemical structure and the data-driven result from PCA were considered. Similar models were also used to evaluate the association of sphingolipid scores with MetS risk (see above). In sensitivity analyses, longitudinal sphingolipid-MetS associations were examined among participants without taking lipid-lowering medication. The RRs (95% CIs) for incident MetS were also calculated according to joint classification of sphingolipid scores and age, sex, obesity status, or cytokine index. *p*_interaction_ was calculated using a likelihood ratio test.

All analyses were performed with R (version 3.5.3) and SAS 9.4 (SAS Institute, Cary, NC, USA). A two-sided *p* value < 0.05 was considered statistically significant, unless specified otherwise.

## 3. Results

### 3.1. Baseline Characteristics

At 6-year resurvey, 431 of 1242 (34.7%) participants developed MetS. Compared with noncases, cases were more likely to be female, urban residents, and have a family history of chronic diseases such as CVD and diabetes at baseline. They also exhibited higher levels of BMI, waist circumference, blood pressure, insulin, HOMA-IR, LDL-cholesterol, triglycerides, triglycerides/HDL-cholesterol, CRP, and RBP4, as well as less optimal levels of HDL-cholesterol, HDL-cholesterol/LDL-cholesterol, and adiponectin (all *p* < 0.001; Table 1). Appendix A presents the differences of sphingolipid concentrations between cases and noncases.

Most of ceramides, dhCers, and SMs showed unfavorable correlations with obesity (*r_s_* = 0.06 to 0.15), triglycerides/HDL-cholesterol (*r_s_* = 0.06 to 0.28), CRP (*r_s_* = 0.06 to 0.17), IL-6 (*r_s_* = 0.07 to 0.13), adiponectin (*r_s_* = −0.06 to −0.14), and RBP4 (*r_s_* = 0.06 to 0.24). In contrast, SM (OH)s, SM (2OH)s, and GSLs exhibited favorable correlations with obesity (*r_s_* = −0.07 to −0.19), triglycerides/HDL-cholesterol (*r_s_* = −0.08 to −0.20), CRP (*r_s_* = −0.08 to −0.14), adiponectin (*r_s_* = 0.06 to 0.15), and RBP4 (*r_s_* = −0.06 to −0.15) (Appendix A).

### 3.2. Factor Analysis

After multivariable adjustment for conventional risk factors, positive associations for incident MetS were detected in eight sphingolipids [five ceramides, two SMs, and SM (2OH) C42:4] with very-long-chain acyl chains (≥20 carbons) (RR_per SD_: 1.10–1.16); whereas the inverse associations were observed in 13 sphingolipids [nine SM (OH)s, two SMs (C34:0, C34:2), SM (2OH) C32:1, and ceramide (d18:1/14:0)] (RR_per SD_: 0.83–0.90) (all FDR-corrected *p* < 0.05; Figure 1).

Because individual sphingolipid species are highly correlated (*r_s_* ranging from −0.37 to 0.98; Appendix A), PCA was performed to simultaneously study the association between 76 sphingolipids and MetS incidence (Appendix A). Of the seven factors identified, Factor 3 (ceramides and dhCers) and Factor 7 (very-long-chain ceramides and dhCers) were positively associated and Factor 1 [mainly containing SM (OH)s] was inversely associated with incident MetS (Table 2, model 1). After further controlling for baseline triglycerides, LDL-cholesterol, and HDL-cholesterol, these associations were attenuated but remained significant (Table 2, model 2). Comparing the extreme quartiles, the RRs (95% CIs) for MetS incidence were 0.66 (0.50, 0.89) (*p*_trend_ = 0.004) for Factor 1, 1.49 (1.19, 1.87) (*p*_trend_ = 0.001) for Factor 3, and 1.43 (1.13, 1.79) (*p*_trend_ < 0.001) for Factor 7. Additional adjustment for inflammatory markers and adipokines had little effect on it (Table 2, model 3). Factor 1 was also inversely associated with reduced HDL-cholesterol and elevated blood pressure, while Factor 3 was positively associated with hypertriglyceridemia and Factor 7 was positively associated with hypertriglyceridemia as well as reduced HDL-cholesterol (Appendix A).

### 3.3. Sphingolipid Scores and Incident MetS Risk

Based on these sphingolipid patterns, five sphingolipid scores were further created according to their chemical structures. As shown in Table 3, elevated ceramide score was associated with a higher MetS risk (RR_Q4 versus Q1_ 1.34; 95% CI 1.06, 1.70; *p*_trend_ = 0.010), whereas increased SM (OH) score was associated with a lower MetS risk (RR_Q4 versus Q1_ 0.71; 95% CI 0.51, 0.97; *p*_trend_ = 0.018), after multivariable adjustment including BMI, baseline triglycerides, LDL-cholesterol, and HDL-cholesterol. These significant associations did not change materially with additional control for inflammatory markers and adipokines. In a sensitivity analysis for MetS, the observed associations were not altered substantially after all participants taking lipid-lowering medication were excluded (Appendix A, model 2). Other sphingolipid scores were not significantly associated with MetS incidence. Among the components of MetS, elevated ceramide score was also associated with increased risk of hypertriglyceridemia (RR_per SD_ 1.15; 95% CI 1.06, 1.25; *p* = 0.001), while the increased SM (OH) score was significantly associated with decreased risk of low HDL-cholesterol (RR_per SD_ 0.81; 95% CI 0.67, 0.97; *p* = 0.019) and elevated blood pressure (RR_per SD_ 0.87; 95% CI 0.77, 0.99; *p* = 0.041) (Appendix A).

### 3.4. Joint Associations of Sphingolipid Scores, Inflammatory Markers, Adipokines, and Incident MetS

The ceramide score showed consistent and positive correlations with CRP (*r_s_* = 0.14; *p* < 0.001) and IL-6 (*r_s_* = 0.10; *p* < 0.001), but inverse correlation with adiponectin (*r_s_* = −0.10; *p* < 0.001) (Figure 2). Using the median of inflammatory index and adipokine index as the cutoff points, the association between ceramide score and MetS incidence was more pronounced in participants with higher inflammation index (RR_Q4 versus Q1_ 1.57; 95% CI 1.16, 2.12; *p*_trend_ = 0.002) than those with lower inflammation index (RR_Q4 versus Q1_ 1.20; 95% CI 0.79, 1.82; *p*_trend_ = 0.336). A significant interaction was observed between ceramide score and inflammation status (*p*_interaction_ = 0.004; Table 4). Consistently, significant modification effects of CRP and IL-6 on the ceramide-MetS association were also detected (Appendix A). No significant interactions were observed when the analysis was stratified according to age, sex, BMI, adipokine index (*p*_interaction_ > 0.05).

## 4. Discussion

Using a high-coverage targeted lipidomic approach, we identified several individual molecular species and two sphingolipid subclasses prospectively associated with MetS risk. Baseline ceramide score was positively associated with incident MetS, whereas baseline SM (OH) score was inversely associated with MetS incidence, after adjustment for socioeconomic information, lifestyle characteristics, and BMI. With additional adjustment for clinical lipids, these associations were attenuated but remained significant. Moreover, a stronger association between ceramide score and risk of MetS was evidenced in individuals with higher inflammation levels than their counterparts.

To the best of our knowledge, this is the first prospective study to comprehensively investigate the associations between plasma sphingolipid concentrations and future risk of MetS. Previously, only a handful of case–control and cross-sectional studies conducted in the U.S. and European populations have investigated the relationships of ≤21 sphingolipids with MetS risk [13,14,15]. For example, by measuring 18 sphingolipids in 76 French men, Denimal et al. [14] reported that five ceramides (d18:1/16:0, d18:1/18:0, d18:1/20:0, d18:1/22:0, d18:1/24:0) and three SMs (d18:1/18:0, d18:1/18:1, d18:1/20:0) were significantly higher in obese individuals with MetS than those in normal controls. More recently, Yin and colleagues [29] evaluated the associations of 30 sphingolipid metabolites with longitudinal changes of metabolic risk factors among 658 participants in the Framingham Heart Study. They found that baseline ceramide(d18:1/24:1) was positively associated with triglyceride change, while saturated SMs such as SM C36:0 and SM(d18:0/24:0) were positively associated with glucose change. With a prospective design and a wider coverage of 76 sphingolipids, our study provided a unique opportunity to discover early metabolic perturbations preceding the clinically identified MetS, as well as laid the groundwork for future studies to explore their potential therapeutic applications.

Among the lipids of interest, several ceramides displayed significant positive associations with MetS incidence, which was in agreement with earlier prospective studies reporting that ceramide species including C18:0, C20:0, and C24:1 were positively associated with CVD [7,30] and T2D [12,31] in Western populations. Our study also extended these findings to include far more monounsaturated ceramide species (C20:1, C22:1). It is worthy to note that these ceramides were independently associated with incident hypertriglyceridemia, suggesting that ceramides may exert additional adverse effects on both the development of hypertriglyceridemia and its progression to MetS. Indeed, compelling evidence from animal models has shown that ceramides could aggregate and retain triglyceride-rich lipoprotein in the vessel wall by regulating expression of sterol regulatory element-binding proteins, fatty acid translocases like CD36, and hormone-sensitive lipase [32]. Additionally, ceramides might also elicit insulin resistance, adverse adipokine release, and mitochondrial stress, leading to metabolic disorders [20]. Overall, our study provides the first evidence that the accumulation of ceramides is related to metabolic disorders in Asian populations. Given that the detrimental effect of ceramides on cardiometabolic risk might be mitigated by lifestyle [33] and diet (e.g., Mediterranean Diet) intervention [30], it might be an effective strategy for clinically targeting ceramides for MetS prevention.

Another novel and interesting finding was that elevated SM (OH) levels were associated with reduced MetS risk, as well as reduced risks of MetS components including low HDL-cholesterol and elevated blood pressure. SM (OH)s are a subclass of low-abundant but common SMs, which require fatty acid hydroxylase 2 (FA2H) in their synthesis [34]. Previously, the favorable relationships of SM (OH)s with cardiometabolic outcomes were reported in two cross-sectional studies and one prospective study [35,36,37]. For instance, an inverse association between SM (OH) C32:2 and incident T2D was reported in the European Prospective Investigation into Cancer and Nutrition–Potsdam cohort [35]. It is worthy to note that, unlike other SMs having atherogenic tendency [38], we found that higher plasma SM (OH)s were associated with better lipid profiles. Compared with other SM subclasses, SM (OH)s displayed stronger positive correlation with HDL-cholesterol but null with triglycerides. Though the exact mechanisms linking SM (OH)s and cardiometabolic health remain largely unclear, lowering SM (OH) levels by knockdown FA2H in 3T3-L1 adipocytes enhanced lipogenesis and decreased glucose uptake by promoting diffusional mobility of raft-associated lipids [39]. Moreover, decreased total SM content in HDL-cholesterol could enhance activity of both hepatic lipase and lecithin-cholesterol acyl transferase, resulting in poor cholesterol efflux and augmented MetS in human studies [14]. Collectively, our study highlighted that potential cardiovascular benefit of SM (OH)s and distinct structures of these lipid metabolites might lead to different health outcomes.

Surprisingly, we found that the detrimental association between ceramide score and incident MetS was more pronounced in participants with higher inflammation levels. This finding was partially supported by the reported stronger association of Cer(d18:1/18:0) with major adverse cardiovascular event among participants with higher CRP level (RR_Q4 versus Q1_: 10.1 versus 2.6) in a Finnish cohort study [40]. The modulatory effect of inflammation on the ceramide-MetS association might be ascribed to the fact that chronic inflammation could directly alter ceramide metabolism. As indicated by animal studies, ceramide biosynthesis can be enhanced when exposing to inflammatory agents such as agonists of toll-like receptors and interleukins [32]. On the other hand, chronic low-grade inflammation is a well-established risk factor for MetS or its features like dyslipidemia in numerous studies including our previous ones from the current cohort population [18,19]. Thus, increased levels of inflammatory factors and ceramide score in our study might additively heighten the risk of incident MetS. In fact, inflammatory cytokines like IL-6 were shown to link with dyslipidemia by promoting de novo hepatic fatty acid synthesis and suppressing the activity of lipoprotein lipase, a key enzyme involving in catabolism of triglyceride-rich lipoproteins [41]. Of notice, although there were modest correlations between ceramide score and levels of CRP and IL-6, controlling for these factors did not materially change the significant ceramide-MetS association, implying that the ceramide-associated metabolic disorder might not fully overlap with inflammation-mediated pathways. Taken together, our findings suggested that combining ceramide score with inflammatory status might facilitate identifying persons with high future cardiometabolic risks.

In this well-characterized cohort study, we have carefully collected and adjusted for multiple potential covariates to minimize confounding. High-coverage targeted lipidomics allows the accurate quantitation of sphingolipid profile and systematic evaluation of their relationships with MetS risk. However, there were some limitations. First, only 76.9% participants were successfully followed up in the 6-year resurvey, although similar rates were reported in other cohort studies [42,43]. Notably, individuals lost to follow-up were more likely to be urban residents, and to have higher levels of educational attainment, family income, and plasma triglycerides [22]. Second, the sphingolipid concentrations were measured only at baseline; thus, we were unable to investigate the association between changes of sphingolipid and MetS risk. Third, our findings, especially for the observed modification effect between inflammation factors and ceramide score, warrant replication in another independent population. Fourth, we cannot establish causality because of the observational nature. Last, the current study population was limited to the middle-aged and older Chinese individuals, and the findings may not be generalized to populations with different ages and ethnic backgrounds.

## 5. Conclusions

In summary, our study indicated that elevated baseline ceramide score was associated with a higher MetS risk, especially among individuals with higher inflammation levels, whereas raised baseline SM (OH) score was associated with a lower MetS risk. These findings suggest that specific sphingolipid molecules might serve as promising novel clinical biomarkers for metabolic risk evaluation and also provide new insights to advance our understanding about relevant etiology, as well as novel intervention strategies for MetS prevention via modifying targeted sphingolipid(s). Certainly, future studies are merited to confirm our findings and also to uncover the underlying mechanisms.

## Figures and Tables

**Figure 1 nutrients-13-02263-f001:**
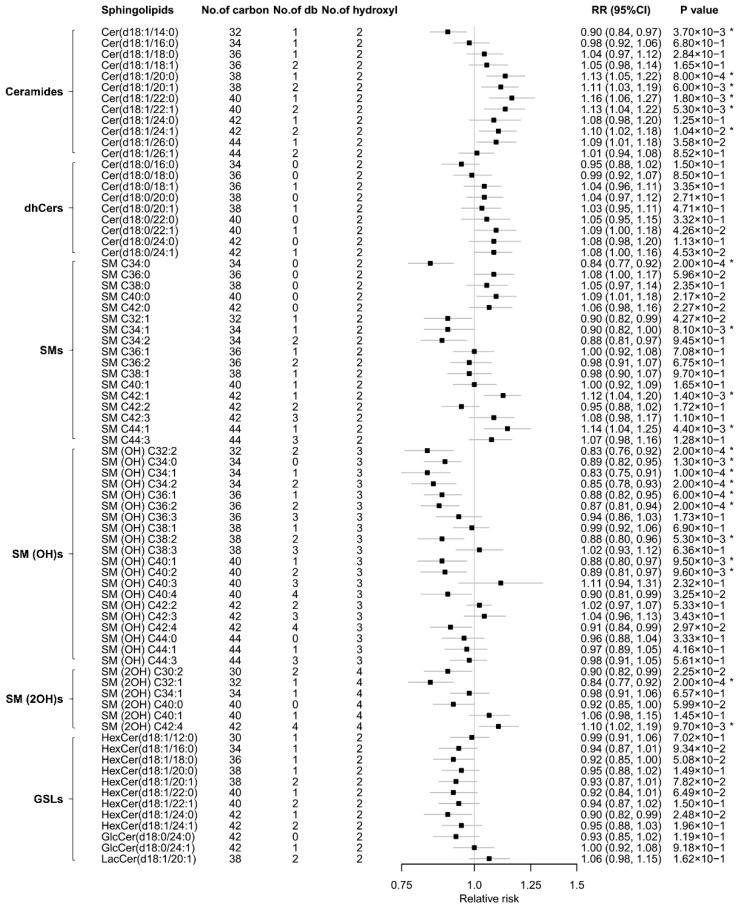
RRs (95% CIs) of individual sphingolipids with incident metabolic syndrome. Model was adjusted for age, sex, region (Beijing or Shanghai), residence (urban or rural), educational attainment (0–6 years, 7–9 years, or ≥10 years), current smoking (yes or no), current alcohol drinking (yes or no), physical activity (low, moderate, or high), family history of chronic diseases (yes or no), use of lipid-lowering medication (yes or no), and BMI. *, *p* < 0.05 after correction for multiple testing using the Benjamini–Hochberg method. Cer, ceramide; db, double-bound; dhCer, dihydroceramide; GlcCer, glucosylceramide; GSL, glycosphingolipid; HexCer, hexosylceramide; LacCer, lactosylceramide; No, number; RR, relative risk; SM, sphingomyelin; SM (OH), sphingomyelin with one additional hydroxyl; SM (2OH), sphingomyelin with two additional hydroxyls.

**Figure 2 nutrients-13-02263-f002:**
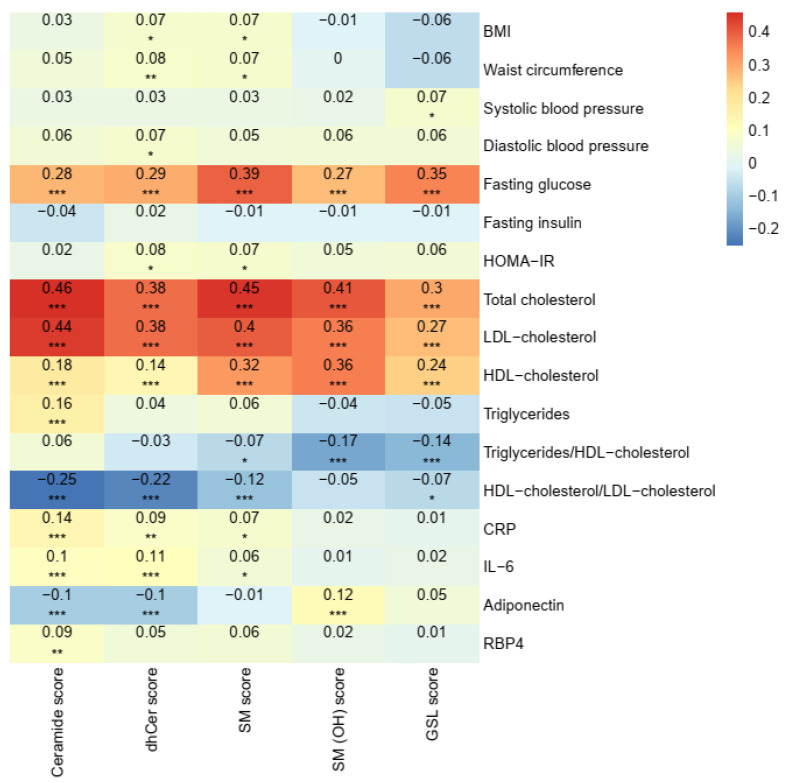
Correlations between sphingolipid scores and metabolic risk factors at baseline. Values were adjusted for age, sex, region (Beijing or Shanghai), and residence (urban or rural). *, *p* < 0.05; **, *p* < 0.01; ***, *p* < 0.001 after correction for multiple testing using the Benjamini–Hochberg method. CRP, C-reactive protein; dhCer, dihydroceramide; GSL, glycosphingolipid; HOMA-IR, homeostatic model assessment of insulin resistance; IL-6, interleukin-6; RBP4, retinol-binding protein 4; SM, sphingomyelin; SM (OH), sphingomyelin with one additional hydroxyl.

**Table 1 nutrients-13-02263-t001:** Baseline characteristics of study population between incident MetS cases and noncases.

	Incident MetS		*p* Value
	No (*n* = 811)	Yes (*n* = 431)	
Age, years	57.9 ± 5.9	57.9 ± 6.0	0.45
Male, *n* (%)	425 (52.4)	162 (37.6)	<0.001
Northern residents, *n* (%)	333 (41.1)	186 (43.2)	0.34
Urban residents, *n* (%)	288 (35.5)	193 (44.8)	0.006
Educational attainment, ≥10 years, *n* (%)	144 (17.8)	92 (21.3)	0.89
Current smoking, *n* (%)	276 (34.0)	109 (25.3)	0.87
Current alcohol drinking, *n* (%)	257 (31.7)	111 (25.8)	0.84
High physical activity, *n* (%)	484 (59.7)	242 (56.1)	0.49
Family history of chronic diseases, *n* (%) ^1^	431 (53.1)	184 (42.7)	0.005
Use of lipid-lowering medication, *n* (%)	10 (1.2)	19 (4.4)	0.002
BMI (kg/m^2^)	22.1 ± 2.6	24.4 ± 3.0	<0.001
Waist circumference (cm)	76.4 ± 8.2	83.0 ± 8.8	<0.001
Systolic blood pressure (mmHg)	130.6 ± 20.8	137.4 ± 20.3	<0.001
Diastolic blood pressure (mmHg)	76.1 ± 10.1	79.7 ± 10.1	<0.001
Fasting glucose (mmol/L)	5.24 ± 0.54	5.20 ± 0.44	0.43
Fasting insulin (μU/mL) ^2^	11.5 (8.5, 15.0)	13.3 (9.7, 17.8)	<0.001
HOMA-IR ^2^	2.63 (1.95, 3.47)	3.09 (2.18, 4.12)	<0.001
Total cholesterol (mmol/L)	4.47 ± 0.90	4.66 ± 0.89	0.06
LDL-cholesterol (mmol/L)	3.00 ± 0.90	3.27 ± 0.88	<0.001
HDL-cholesterol (mmol/L)	1.45 ± 0.34	1.32 ± 0.28	<0.001
Triglycerides (mmol/L)	0.78 (0.57, 1.01)	1.07 (0.80, 1.38)	<0.001
Triglycerides/HDL-cholesterol	0.55 (0.37, 0.76)	0.80 (0.56, 1.16)	<0.001
HDL-cholesterol/LDL-cholesterol	0.52 ± 0.20	0.42 ± 0.13	<0.001
CRP (mg/L)	0.39 (0.12, 0.85)	0.59 (0.32, 1.21)	<0.001
IL-6 (pg/mL) ^3^	0.90 (0.57, 1.38)	1.00 (0.62, 1.47)	0.06
Adiponectin (μg/mL) ^3^	17.4 (10.9, 26.6)	13.3 (8.7, 21.4)	<0.001
RBP4 (μg/mL)	35.8 (30.0, 43.0)	38.9 (32.3, 46.3)	<0.001

Values are mean ± SD, median (interquartile range), or *n* (%). Data may not sum to 100 because of rounding. *p* values were calculated adjusted for age, sex, region (Beijing or Shanghai), and residence (urban or rural). CRP, C-reactive protein; HOMA-IR, homeostatic model assessment of insulin resistance; IL-6, interleukin-6; MetS, metabolic syndrome; RBP4, retinol-binding protein 4. ^1^ Includes coronary heart disease, stroke, hypertension, and diabetes in a parent or a first-degree sibling. ^2^ Data are missing for two participants. ^3^ Data are missing for 33 participants.

**Table 2 nutrients-13-02263-t002:** Association between baseline sphingolipid factors (PCA extracted) and incident metabolic syndrome.

	Quartiles of Factors				*p* _trend_	Per SD Increment	*p*
	Q1	Q2	Q3	Q4			
Factor 1 (Hydroxysphingomyelins, long-chain SMs, and GSLs)							
Model 1	1	0.76 (0.62, 0.94)	0.64 (0.50, 0.82)	0.57 (0.44, 0.73)	<0.001	0.80 (0.74, 0.88)	<0.001
Model 2	1	0.81 (0.65, 1.00)	0.72 (0.56, 0.93)	0.66 (0.50, 0.89)	0.004	0.83 (0.76, 0.91)	<0.001
Model 3	1	0.81 (0.65, 1.00)	0.71 (0.55, 0.91)	0.67 (0.51, 0.90)	0.004	0.83 (0.75, 0.92)	<0.001
Factor 2 (SMs)							
Model 1	1	1.36 (1.10, 1.68)	1.30 (1.04, 1.62)	1.20 (0.94, 1.53)	0.26	1.08 (0.99, 1.17)	0.069
Model 2	1	1.23 (1.00, 1.51)	1.17 (0.94, 1.45)	1.05 (0.83, 1.33)	0.996	1.03 (0.95, 1.12)	0.49
Model 3	1	1.14 (0.93, 1.41)	1.08 (0.87, 1.35)	0.98 (0.78, 1.25)	0.65	1.01 (0.92, 1.10)	0.86
Factor 3 (Ceramides and dhCers)							
Model 1	1	1.34 (1.07, 1.68)	1.35 (1.08, 1.70)	1.55 (1.23, 1.94)	<0.001	1.13 (1.05, 1.21)	0.001
Model 2	1	1.37 (1.10, 1.71)	1.34 (1.07, 1.67)	1.49 (1.19, 1.87)	0.001	1.11 (1.03, 1.20)	0.006
Model 3	1	1.36 (1.08, 1.70)	1.32 (1.05, 1.66)	1.50 (1.19, 1.89)	0.002	1.12 (1.04, 1.22)	0.004
Factor 4 (GSLs)							
Model 1	1	1.03 (0.86, 1.23)	0.83 (0.68, 1.02)	0.79 (0.62, 1.01)	0.020	0.90 (0.83, 0.98)	0.011
Model 2	1	1.08 (0.90, 1.29)	0.93 (0.76, 1.14)	0.90 (0.70, 1.15)	0.27	0.96 (0.89, 1.04)	0.36
Model 3	1	1.10 (0.92, 1.32)	0.96 (0.78, 1.18)	0.96 (0.76, 1.22)	0.57	0.99 (0.91, 1.07)	0.79
Factor 5 (Very-long-chain SMs and hydroxysphingomyelins)							
Model 1	1	1.04 (0.84, 1.28)	1.12 (0.91, 1.37)	1.09 (0.88, 1.34)	0.37	1.03 (0.96, 1.12)	0.39
Model 2	1	0.99 (0.81, 1.22)	1.11 (0.90, 1.37)	1.06 (0.84, 1.34)	0.48	1.04 (0.96, 1.13)	0.38
Model 3	1	1.00 (0.81, 1.24)	1.09 (0.88, 1.35)	1.09 (0.86, 1.38)	0.40	1.05 (0.96, 1.14)	0.29
Factor 6 (Very-long-chain hydroxysphingomyelins)							
Model 1	1	1.29 (1.06, 1.57)	1.15 (0.92, 1.44)	1.16 (0.93, 1.44)	0.29	1.03 (0.96, 1.11)	0.39
Model 2	1	1.33 (1.09, 1.61)	1.26 (1.01, 1.57)	1.25 (1.01, 1.56)	0.055	1.06 (0.98, 1.14)	0.15
Model 3	1	1.33 (1.09, 1.62)	1.27 (1.02, 1.58)	1.29 (1.04, 1.62)	0.028	1.06 (0.98, 1.14)	0.15
Factor 7 (Very-long-chain ceramides and dhCers)							
Model 1	1	1.17 (0.92, 1.49)	1.52 (1.21, 1.91)	1.68 (1.34, 2.11)	<0.001	1.23 (1.14, 1.32)	<0.001
Model 2	1	1.12 (0.88, 1.43)	1.43 (1.14, 1.78)	1.43 (1.13, 1.79)	<0.001	1.15 (1.07, 1.24)	<0.001
Model 3	1	1.14 (0.90, 1.45)	1.45 (1.16, 1.82)	1.42 (1.13, 1.80)	<0.001	1.14 (1.06, 1.23)	<0.001

Data are RR (95% CI) from multivariable-adjusted log-Poisson models. dhCer, dihydroceramide; GSL, glycosphingolipid; PCA, principal component analysis; RR, relative risk; SM, sphingomyelin. Model 1, adjusted for age, sex, region (Beijing or Shanghai), residence (urban or rural), educational attainment (0–6 years, 7–9 years, or ≥10 years), current smoking (yes or no), current alcohol drinking (yes or no), physical activity (low, moderate, or high), family history of chronic diseases (yes or no), use of lipid-lowering medication (yes or no), and BMI. Model 2, further adjusted for triglycerides, LDL-cholesterol, and HDL-cholesterol at baseline. Model 3, further adjusted for HOMA-IR, inflammatory markers (C-reactive protein and interleukin-6), and adipokines (adiponectin and retinol-binding protein 4) at baseline.

**Table 3 nutrients-13-02263-t003:** Association between baseline sphingolipid scores and incident metabolic syndrome.

	Quartiles of Scores				*p* _trend_	Per SD Increment	*p*
	Q1	Q2	Q3	Q4			
Ceramide score; *n* of molecules = 10							
Model 1	1	1.17 (0.94, 1.47)	1.31 (1.06, 1.63)	1.31 (1.05, 1.63)	0.010	1.10 (1.02, 1.19)	0.009
Model 2	1	1.17 (0.94, 1.47)	1.33 (1.07, 1.66)	1.34 (1.06, 1.70)	0.010	1.10 (1.02, 1.20)	0.016
Model 3	1	1.14 (0.91, 1.43)	1.30 (1.04, 1.62)	1.33 (1.05, 1.69)	0.013	1.11 (1.02, 1.21)	0.016
dhCer score; *n* of molecules = 8							
Model 1	1	1.16 (0.92, 1.45)	1.12 (0.89, 1.40)	1.16 (0.92, 1.46)	0.27	1.05 (0.97, 1.13)	0.24
Model 2	1	1.17 (0.94, 1.46)	1.16 (0.92, 1.46)	1.11 (0.87, 1.41)	0.51	1.04 (0.96, 1.13)	0.32
Model 3	1	1.15 (0.92, 1.44)	1.15 (0.92, 1.45)	1.10 (0.86, 1.41)	0.50	1.05 (0.96, 1.15)	0.25
SM score; *n* of molecules = 6							
Model 1	1	1.02 (0.83, 1.26)	0.81 (0.65, 1.02)	0.79 (0.63, 1.01)	0.02	0.91 (0.84, 1.00)	0.04
Model 2	1	1.07 (0.88, 1.31)	0.88 (0.69, 1.11)	0.85 (0.65, 1.11)	0.14	0.94 (0.85, 1.03)	0.20
Model 3	1	1.03 (0.85, 1.27)	0.84 (0.66, 1.06)	0.83 (0.64, 1.09)	0.11	0.94 (0.85, 1.03)	0.19
SM (OH) score; *n* of molecules = 11							
Model 1	1	0.82 (0.67, 1.01)	0.62 (0.49, 0.80)	0.60 (0.45, 0.79)	<0.001	0.82 (0.75, 0.91)	<0.001
Model 2	1	0.91 (0.74, 1.13)	0.75 (0.58, 0.98)	0.71 (0.51, 0.97)	0.018	0.87 (0.78, 0.97)	0.011
Model 3	1	0.89 (0.72, 1.10)	0.77 (0.59, 1.00)	0.75 (0.55, 1.04)	0.059	0.88 (0.79, 0.98)	0.026
GSL score; *n* of molecules = 5							
Model 1	1	0.97 (0.80, 1.17)	0.73 (0.58, 0.91)	0.85 (0.67, 1.09)	0.10	0.94 (0.85, 1.02)	0.15
Model 2	1	1.02 (0.84, 1.24)	0.81 (0.65, 1.01)	0.96 (0.74, 1.24)	0.49	0.98 (0.89, 1.08)	0.70
Model 3	1	1.04 (0.85, 1.26)	0.78 (0.63, 0.98)	0.99 (0.77, 1.27)	0.58	0.99 (0.90, 1.09)	0.90

Data are RR (95% CI) from multivariable-adjusted log-Poisson models. dhCer, dihydroceramide; GSL, glycosphingolipid; RR, relative risk; SM, sphingomyelin; SM (OH), sphingomyelin with one additional hydroxyl. Model 1, adjusted for age, sex, region (Beijing or Shanghai), residence (urban or rural), educational attainment (0–6 years, 7–9 years, or ≥10 years), current smoking (yes or no), current alcohol drinking (yes or no), physical activity (low, moderate, or high), family history of chronic diseases (yes or no), use of lipid-lowering medication (yes or no), and BMI. Model 2, further adjusted for triglycerides, LDL-cholesterol, and HDL-cholesterol at baseline. Model 3, further adjusted for HOMA-IR, inflammatory markers (C-reactive protein and interleukin-6), and adipokines (adiponectin and retinol-binding protein 4) at baseline.

**Table 4 nutrients-13-02263-t004:** Stratified analysis of RR (95% CI) of incident metabolic syndrome according to quartiles of baseline sphingolipid scores.

	**Quartile of Ceramide Score**					***p*_trend_**	***p*_interaction_**
	***n***	**Q1**	**Q2**	**Q3**	**Q4**		
Age							0.785
≤59 years	744	1	1.03 (0.77, 1.38)	1.15 (0.86, 1.52)	1.12 (0.83, 1.52)	0.413	
>59 years	498	1	1.13 (0.81, 1.58)	1.40 (1.00, 1.97)	1.40 (0.96, 2.03)	0.042	
Sex							0.260
Men	587	1	1.26 (0.86, 1.84)	1.47 (1.01, 2.15)	1.50 (0.99, 2.25)	0.052	
Women	655	1	1.23 (0.95, 1.58)	1.19 (0.91, 1.56)	1.20 (0.90, 1.60)	0.257	
BMI ^1^							0.578
<25 kg/m^2^	978	1	1.14 (0.84, 1.54)	1.43 (1.08, 1.90)	1.21 (0.87, 1.68)	0.168	
≥25 kg/m^2^	264	1	1.21 (0.88, 1.66)	1.33 (0.97, 1.83)	1.53 (1.11, 2.09)	0.005	
Inflammatory index							0.004
≤median	603	1	1.44 (1.01, 2.05)	1.56 (1.11, 2.20)	1.20 (0.79, 1.82)	0.336	
>median	603	1	1.19 (0.89, 1.59)	1.40 (1.03, 1.88)	1.57 (1.16, 2.12)	0.002	
Adipokine index							0.422
≤median	604	1	1.14 (0.84, 1.53)	1.11 (0.82, 1.51)	1.06 (0.76, 1.47)	0.838	
>median	605	1	1.18 (0.85, 1.66)	1.63 (1.16, 2.29)	1.70 (1.19, 2.44)	<0.001	
	**Quartile of SM (OH) Score**					***p*_trend_**	***p*_interaction_**
	***n***	**Q1**	**Q2**	**Q3**	**Q4**		
Age							0.194
≤59 years	744	1	0.84 (0.64, 1.10)	0.68 (0.49, 0.95)	0.53 (0.36, 0.78)	0.001	
>59 years	498	1	0.88 (0.63, 1.23)	0.77 (0.51, 1.18)	1.14 (0.69, 1.88)	0.898	
Sex							0.048
Men	587	1	0.75 (0.53, 1.06)	0.65 (0.42, 1.00)	0.85 (0.50, 1.42)	0.446	
Women	655	1	1.00 (0.77, 1.30)	0.88 (0.65, 1.21)	0.72 (0.49, 1.06)	0.121	
BMI ^1^							0.434
<25 kg/m^2^	978	1	0.77 (0.57, 1.03)	0.74 (0.53, 1.04)	0.65 (0.43, 0.99)	0.050	
≥25 kg/m^2^	264	1	1.13 (0.83, 1.54)	1.05 (0.74, 1.50)	1.24 (0.82, 1.88)	0.421	
Inflammatory index							0.823
≤median	603	1	0.91 (0.65, 1.26)	0.70 (0.47, 1.06)	0.62 (0.39, 1.01)	0.033	
>median	603	1	0.92 (0.69, 1.23)	0.87 (0.61, 1.25)	0.90 (0.59, 1.38)	0.608	
Adipokine index							0.819
≤median	604	1	0.73 (0.53, 0.99)	0.81 (0.58, 1.13)	0.78 (0.52, 1.18)	0.311	
>median	605	1	0.73 (0.53, 1.00)	0.67 (0.46, 0.97)	0.68 (0.42, 1.10)	0.094	

Model was adjusted for age, sex, region (Beijing or Shanghai), residence (urban or rural), educational attainment (0–6 years, 7–9 years, or ≥10 years), current smoking (yes or no), current alcohol drinking (yes or no), physical activity (low, moderate, or high), family history of chronic diseases (yes or no), use of lipid-lowering medication (yes or no), BMI, triglycerides, LDL-cholesterol, and HDL-cholesterol at baseline. Inflammatory index and adipokine index were computed as follows: (CRP z score + IL-6 score)/2 and (adiponectin z score + RBP4 z score)/2, respectively. ^1^ Modified metabolic syndrome was defined as having two or more components of metabolic syndrome without central obesity.

## Data Availability

The data used during the current study are available on reasonable request from the corresponding author.

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
