# Peer review of "Plasma Sphingolipid Profile in Association with Incident Metabolic Syndrome in a Chinese Population-Based Cohort Study"

_nutrients, 2021, doi:10.3390/nu13072263_

Round 1

Reviewer 1 Report

The manuscript shows the results of an observational longitudinal study (Nutrition and Health of Aging People in China Study) of 1242 Chinese patients aged 50 to 70 years with a 6-year follow-up. Participants were assessed at baseline with the dosing of 76 sphingolipids and other blood and anthropometric markers related to the metabolic syndrome. The included participants did not have clear evidence of metabolic syndrome at baseline. At the end of the follow-up, different correlations were evaluated between the classes of plasma sphingolipids detected at baseline and the incidence of metabolic syndrome, applying correction models for metabolic, inflammatory and blood lipid markers. The main findings consist of the positive association between ceramides, dihydroxyceremides and ceramide score with the incidence of metabolic syndrome. Conversely, an inverse correlation was found between hydroxysphingomyelins, glycosphingolipids and hydroxysphingomyelin score and incidence of metabolic syndrome. Furthermore, the highest quartile of ceramide score showed significant association with higher-than-mean inflammatory index, compared to the lowest quartile. This significant interaction highlighted the role of ceramides on inflammatory mechanisms.

The manuscript is very smooth and well written. Bibliographic references are updated and adapted to the topic. Although the study does not explore an innovative aspect, it shows many analytical and statistical evaluations that enrich the available literature data. As would be expected from the type of study, there are several inherent limitations derived from the observational nature, however, the manuscript highlights possible mechanisms of action of sphingolipids that could stimulate future work. Lipidomics is a branch of "omics" sciences that is developing interesting implications for health, especially with the advent of increasingly affordable analyzes.

I would like to suggest some changes/clarifications to the authors. Here are my comments:

- Regarding the association between ceramide score and SM(OH) score with the incidence of metabolic syndrome, statistical model 3 is not sufficiently discussed and the authors focus only on model 1 and 2. However, the inclusion in model 3 of inflammatory markers seems very important, considering the interactions highlighted. Statistical significance should also be discussed.

I advise the authors to discuss the possible clinical implications of the findings in more detail, considering that an analysis of this type could be very expensive and therefore not easy to insert routine investigations. How can this correlation change the clinical approach in the prevention and treatment of metabolic syndrome?

- For greater clarity, it would be more immediate for the reader to follow the participant selection and follow-up process through a flowchart showing the details of the participants. This would allow table S1 to be omitted.

- It is unclear what the authors mean when they state in line 87 that no participant included had relevant components for metabolic syndrome at baseline. Table S1 shows that over 50% of the participants had at least one element of the criteria for metabolic syndrome at baseline. This would imply that the correlations highlighted would only be valid for a Chinese population at risk of metabolic syndrome

- At lines 88-90, the sentence should be rephrased. Again, the flowchart would be able to clarify better. Which evaluations do the authors refer to?

- The abbreviation GSLs at line 126 should also be indicated in extenso as glycosphingolipids, being the first time it was mentioned in the manuscript

- What evaluations and with what frequency were made in the patients during the follow-up?

- Among the limitations of the study it should be noted that the associations are only with the baseline measurements of sphingolipids

Reviewer 2 Report

Plasma sphingolipid profile in association with incident meta- 2 bolic syndrome in a Chinese population-based cohort study

 Huan Yun et al aimed to examine the associations 75 of sphingolipids with risk of incident MetS and to explore potential modifications of in- 76 flammatory markers and adipokines on the associations in a Chinese prospective cohort.

This is an interesting and well written paper. The results are clearly stated and the discussion is thorough.

There are however, some points I want to highlight.

The objective should be clearly stated as this is an epidemiologic study. I think it is necessary to indicate with whom, when and where this study has been carried out. Furthermore, “to examine the ….” Must be changed with “estimate the associations..”

Please change through the paper “multivariate” for multivariable

Please change through the paper P for p

L144: What does logistic regression for categorical variables….incident MetS mean?

There are some typos

Round 2

Reviewer 1 Report

the authors have substantially improved the manuscript